**Subject Category:**
Biology (whole organism)

ecology

stable isotopes, elasmobranch, mobulid, foraging, nutrient transport

**Author for correspondence:**
Lauren R. Peel
e-mail: lauren.peel@research.uwa.edu.au

# Stable isotope analyses reveal unique trophic role of reef manta rays (*Mobula alfredi*) at a remote coral reef

Lauren R. Peel[1,2,3,4], Ryan Daly[3,5], Clare A. Keating Daly[3], Guy M. W. Stevens[4], Shaun P. Collin[1,6] and Mark G. Meekan[2]

[1]School of Biological Sciences, The Oceans Graduate School, The University of Western Australia, Crawley, Western Australia 6009, Australia
[2]The Australian Institute of Marine Science, Crawley, Western Australia 6009, Australia
[3]Save Our Seas Foundation – D'Arros Research Centre (SOSF-DRC), Rue Philippe Plantamour 20, 1201 Genève, Switzerland
[4]The Manta Trust, Catemwood House, Norwood Lane, Corscombe, Dorset DT2 0NT, UK
[5]South African Institute for Aquatic Biodiversity (SAIAB), Private Bag 1015, Grahamstown 6140, South Africa
[6]School of Life Sciences, La Trobe University, Bundoora, Victoria 3086, Australia

LRP, 0000-0001-6960-5663

Stable isotope analyses provide the means to examine the trophic role of animals in complex food webs. Here, we used stable isotope analyses to characterize the feeding ecology of reef manta rays (*Mobula alfredi*) at a remote coral reef in the Western Indian Ocean. Muscle samples of *M. alfredi* were collected from D'Arros Island and St. Joseph Atoll, Republic of Seychelles, in November 2016 and 2017. Prior to analysis, lipid and urea extraction procedures were tested on freeze-dried muscle tissue in order to standardize sample treatment protocols for *M. alfredi*. The lipid extraction procedure was effective at removing both lipids and urea from samples and should be used in future studies of the trophic ecology of this species. The isotopic signatures of nitrogen ($\delta^{15}$N) and carbon ($\delta^{13}$C) for *M. alfredi* differed by year, but did not vary by sex or life stage, suggesting that all individuals occupy the same trophic niche at this coral reef. Furthermore, the isotopic signatures for *M. alfredi* differed to those for co-occurring planktivorous fish species also sampled at D'Arros Island and St. Joseph Atoll, suggesting that the ecological niche of *M. alfredi* is unique. Pelagic zooplankton were the main contributor (45%) to the diet of *M. alfredi*, combined with emergent zooplankton (38%) and mesopelagic prey items (17%). Given the extent of movement that would be required to undertake this foraging strategy, individual

*M. alfredi* are implicated as important vectors of nutrient supply around and to the coral reefs surrounding D'Arros Island and St. Joseph Atoll, particularly where substantial site fidelity is displayed by these large elasmobranchs.

## 1. Background

Coral reefs support high levels of marine biodiversity and host intricate food webs [1,2]. Many reef systems are isolated, dispersed across tropical waters where they form hotspots of increased productivity in otherwise oligotrophic oceans [3]. Part of this productivity may be supported by highly mobile marine megafauna, such as sharks, rays, seabirds, turtles and whales, whose foraging movements and residency patterns may facilitate significant nutrient transport and recycling between reef and offshore environments [4–7].

Reef manta rays (*Mobula alfredi*) [8,9] are large filter-feeding elasmobranchs that display high levels of site fidelity and residency at circum-tropical reef locations [10–13]. Individuals are often observed feeding on pelagic zooplankton that accumulates near the surface of the water column (less than 5 m) during daylight hours, and this foraging behaviour has been found to be linked to zooplankton density in eastern Australia [14]. During the night, demersal zooplankton emerge from the seabed, where they vertically migrate towards the surface [15], and become potential prey items for *M. alfredi* [16]. These emergent zooplankton communities are thought to be particularly significant for *M. alfredi* that occupy lagoon systems [17]. Furthermore, *M. alfredi* have been observed to travel offshore to feed on mesopelagic zooplankton before returning to inshore coral reefs during the day where they may excrete waste products [16,18–20]. In this way, *M. alfredi* may be able to create links between shallow coral reefs and deeper water ecosystems, potentially facilitating the horizontal transport of nutrients between these environments [21].

Stable isotope analyses provide a means to examine the trophic role of manta rays and other marine megafauna in coral reef environments [22–24]. The isotopic ratios of nitrogen ($^{14}N/^{15}N$, or $\delta^{15}N$) and carbon ($^{12}C/^{13}C$, or $\delta^{13}C$) in the muscle tissues of manta rays provide information on both the trophic level and foraging locations of these animals. This is possible as values of $\delta^{15}N$ increase with increasing trophic position [25,26], and values of $\delta^{13}C$ display predictable changes with foraging habitat [27] and location [28]. Few studies to date have used this analytical approach to examine the feeding ecology of reef and oceanic (*Mobula birostris*) manta rays, but those published have shown that emergent ($\delta^{13}C > −17‰$) [27] and mesopelagic zooplankton (greater than 200 m depth in the water column) comprise a significant proportion of the diet of *M. alfredi* along the coast of eastern Australia [16], and of *M. birostris* in Ecuador [29], respectively. Emergent zooplankton have also been reported to be a significant contributor to the diet of *M. alfredi* within the lagoon of Palmyra Atoll in the central Pacific [17]. Coupled with the potential for *M. alfredi* to travel large distances (greater than 300 km) [19,30,31], these findings suggest that manta rays may act as a vector for the horizontal transport of nutrients between offshore and coastal reef ecosystems. Additionally, the high site fidelity displayed by *M. alfredi* at aggregation sites may serve to increase the significance of such nutrient transfer processes, and of the trophic role of this species within reef environments as a whole.

Although previous research suggests that manta rays may be important to nutrient flows in oligotrophic seas [16,17], the context of their trophic role within reef communities is not fully understood as a result of the restricted sampling regimes [32]. Since most studies of stable isotopes have only sampled the target species and putative species of prey [16,29,33], it may be that manta rays are simply one species of a much larger guild of fishes that perform similar functions. Alternatively, by moving across habitats over larger distances than most other planktivorous reef fishes [34], it could be that manta rays occupy a unique role in nutrient transport in reef systems. Insight into this issue requires contemporaneous sampling and isotope analysis across a wide range of species from multiple guilds of reef fishes.

Here, we describe the feeding ecology and trophic role of *M. alfredi* at the coral reefs surrounding D'Arros Island and St. Joseph Atoll (hereafter, D'Arros Island), Republic of Seychelles, using stable isotope analyses. In order to facilitate comparisons across species at D'Arros Island and among studies in other locations [32], we firstly optimized sample treatment procedures by assessing the effect of lipid and urea extraction procedures on the $\delta^{15}N$ and $\delta^{13}C$ values obtained from *M. alfredi* muscle tissue. Our study then assessed the extent to which foraging *M. alfredi* targeted pelagic, emergent and mesopelagic zooplankton communities, given the findings of earlier work [16,27,29].

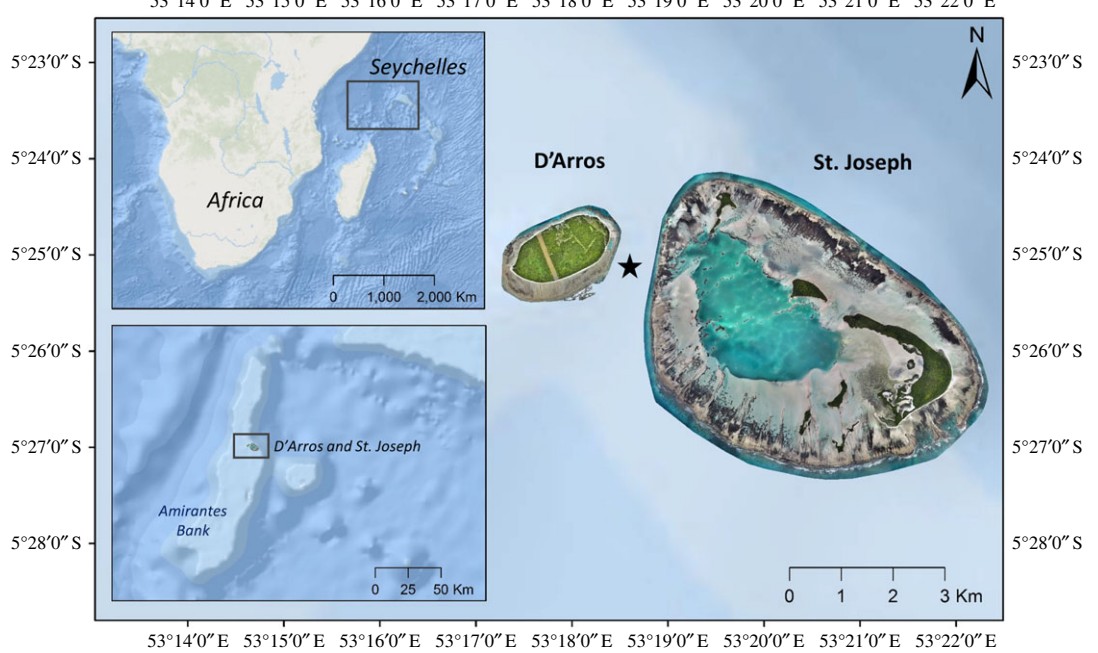

**Figure 1.** D'Arros Island and St. Joseph Atoll, located on the Amirantes Bank within the Republic of Seychelles, Western Indian Ocean. Position of St. Joseph Channel indicated by star. Maps created in ArcGIS 10.3 (http://www.esri.com/) using GEBCO_08 (version 20100927) bathymetry data. Georeferenced drone imagery © Save Our Seas Foundation.

The values of $\delta^{15}N$ and $\delta^{13}C$, and ratios of C : N, for *M. alfredi* were then compared to that of zooplankton, reef fishes and seagrass samples collected at D'Arros Island to understand the broader trophic role of the species within the reef community. Lastly, we examined the role that this species may play in the horizontal transport of nutrients across ecosystems within this region of the Western Indian Ocean.

## 2. Methods

### 2.1. Study site

The Republic of Seychelles is an archipelago located in the Western Indian Ocean and encompasses an Exclusive Economic Zone (EEZ) of 1.4 million $km^2$. It comprises 115 islands that collectively occupy 459 $km^2$ of land [35]. These tropical islands are divided into two main groups based on their geography and composition; the granitic islands to the north comprise the Inner Islands, and the dispersed coralline islands to the southwest comprise the Outer Islands. The Amirante Island Group, located upon the predominantly shallow (less than 40 m depth) Amirantes Bank, lies within the Outer Island region and is made up of 11 low-lying sand cays [36]. D'Arros Island (1.71 $km^2$) and the St. Joseph Atoll (1.63 $km^2$) occur in the central region of the Bank (5°24.9′ S, 53°17.9′ E), and are separated by a 1 km wide and 60 m deep channel (figure 1) [36]. Aggregations of *M. alfredi* are observed year-round in the waters surrounding D'Arros Island [37], which is significant given the infrequency of sightings of this species at other islands and island groups throughout the rest of Seychelles (L. Peel 2018, unpublished data).

### 2.2. Sample collection

#### 2.2.1. Reef manta rays

Small tissue samples (approx. 50 mg; *n* = 50) were collected from the posterior dorsal surface of *M. alfredi* in November 2016 (*n* = 13) and November 2017 (*n* = 37) using a biopsy probe mounted onto the end of a modified Hawaiian sling [16]. Individual *M. alfredi* were identified by the unique pigmentation patterns present on their ventral surface [38,39] prior to all sampling to ensure that the sex, size and identification number of each animal could be associated with all collected samples and to minimize the likelihood of re-sampling the same individual. Sex was determined from the presence (male) or the absence (female) of

claspers [8,40]. Wingspan (m) was visually estimated to the nearest 0.1 m, and individuals subsequently categorized into one of three life-stage classes indicative of increasing maturity status [41]: juvenile (less than or equal to 2.4 m), sub-adult (male, 2.5–2.8 m; female, 2.5–3.1 m), and adult (male, greater than or equal to 2.9 m; female, greater than or equal to 3.2 m). The presence of mating scars on females and the extent of calcification of the claspers of males were used as additional aids to assess maturity status [42,43]. Biopsy samples were kept on ice until return to land, where the white muscle of the sample was separated from the skin, and tissues were stored at −20°C. A single sample of M. alfredi faeces was also collected opportunistically from a surface-feeding individual at D'Arros Island, and subsequently stored at −20°C.

### 2.2.2. Reef fishes

Samples of the dorsal musculature of 20 fish species, representative of nine trophic guilds ranging from herbivore to carnivore, were collected at D'Arros Island in November 2017 (n = 157; electronic supplementary material, table S1). Species were assigned to trophic guilds based on information about their primary diet obtained from FishBase [44]. Capture method, mean size (fork length, cm) and diet of each species are summarized in electronic supplementary material, table S1. Tissue samples of white muscle collected in the field were kept on ice until being stored at −20°C.

### 2.2.3. Zooplankton

Zooplankton samples ($n_{total}$ = 24) were collected from near the surface of the water column around D'Arros Island during the day and at night.

#### 2.2.3.1. Pelagic

During the day, zooplankton samples were collected within the uppermost 2 m of the water column using a small plankton net towed behind an 18 ft research boat. The net (202 µm mesh, 50 cm diameter; General Oceanics, FL, USA) was deployed when M. alfredi were sighted feeding over the reef flats of D'Arros Island or along the reef edge of the St. Joseph Channel during November 2016 (n = 7) and 2017 (n = 10), and towed for approximately 5 min at a speed of 2 knots. The sample contained in the cod-end at the completion of each tow was kept on ice until it was divided into subsamples using a Folsom plankton sample splitter (Aquatic Research Instruments, ID, USA), and stored at −20°C.

#### 2.2.3.2. Emergent

Samples of emergent zooplankton were collected at night using a small light trap constructed from a 300 ml plastic bottle, 200 µm mesh net and an underwater fishing lure light with a white LED in November 2017 (n = 7). The light trap was deployed 40 m offshore to the north of D'Arros Island at a depth of 2 m at sunset, and retrieved after approximately 2.5 h. The collected zooplankton sample was immediately stored at −20°C.

### 2.2.4. Seagrass

Seagrass samples (Thalassodendron ciliatum, n = 10) were collected by hand within the St. Joseph Atoll lagoon in November 2017. Seagrass leaves were removed from the stems and all epiphytes removed from the exterior of the blades prior to storing the samples at −20°C.

## 2.3. Sample processing

Within a month of collection, all samples were lyophilized in an Alpha 1–2 LD Plus freezer dryer (Martin Christ, Germany) for 69.8 ± 14.8 h and subsequently stored in a desiccator until required. In April 2018, all fish, zooplankton and seagrass samples were coarsely subdivided and homogenized by hand before being ground to a fine powder using a Mixed Mill MM 200 with 6.4 mm ball bearings in preparation for stable isotope analysis. The single sample of M. alfredi faeces was also processed in this manner. No additional extraction procedures were performed on this subset of samples.

Freeze-dried tissue samples of M. alfredi were subdivided into 1 × 1 mm cubes by hand using a scalpel blade, but were not ground to a powder because of the sponge-like nature of the freeze-dried tissue.

Given the recommendations made in previous studies of stable isotope ratios in elasmobranch tissues [45–47], lipids and urea were extracted from *M. alfredi* samples following the methodology of Carlisle *et al.* [48] and Marcus *et al.* [47], respectively. Briefly, subdivided samples were soaked in 2 : 1 chloroform-methanol solution for 24 h to remove lipids. After a 24 h air-drying period, the samples were then oven dried for 48 h at 60°C. To remove urea, samples were then soaked in 1.5–1.8 ml of milliQ water for 72 h; centrifuging each sample (Centrifuge 5810 R; Eppendorf, Hamburg, Germany) for 3 min at 3000 r.p.m. and replacing the water every 24 h. At the completion of this process, samples were oven-dried a final time for 48 h at 60°C.

To examine the effect of the lipid and urea extraction on *M. alfredi* muscle tissue, five samples that were collected in 2017—four male and one female—were subdivided into quarters. One of four extraction treatments was then applied to each subsample. The first subsample was exposed to the full extraction treatment described above (lipid and urea extraction; LE + DIW), the second was exposed to only the lipid extraction treatment (LE) and the third to only the urea extraction treatment (DIW). The last subsample was left untreated as a control according to Marcus *et al.* [47].

## 2.4. Stable isotope analysis

All samples were analysed for $\delta^{15}N$ and $\delta^{13}C$, using a continuous flow system consisting of a Delta V Plus mass spectrometer connected with a Thermo Flush 1112 via Conflo IV (Thermo-Finnigan, Germany) at the West Australian Biogeochemistry Centre at The University of Western Australia. $\delta^{15}N$ and $\delta^{13}C$ (parts per million; ‰) were used to express stable isotope ratios, with $\delta^{15}N$ reported relative to atmospheric $N_2$ and $\delta^{13}C$ reported relative to the standard reference Vienna Pee Dee Belemnite. Samples were standardized against primary analytical standards from the International Atomic Energy Agency ($\delta^{13}C$: NBS22, USGS24, NBS19, LSVEC; $\delta^{15}N$: N1, N2, USGS32 and laboratory standards). The external error of analyses, calculated as the standard deviation of mean values, was determined to be 0.10‰ for both $\delta^{15}N$ and $\delta^{13}C$.

Prior to data analysis, lipid normalization equations were applied where the reported mean C : N ratios for sampled fauna were greater than 3.5 [49] as the presence of lipids in muscle tissue can lead to depleted values of $\delta^{13}C$ [50,51]. No corrections were required for *M. alfredi* or any of the reef fishes, but zooplankton $\delta^{13}C$ values were normalized with the following equation [29,52]:

$$\delta^{13}C_{norm} = \delta^{13}C_{bulk} + 7.95 \times \left( \frac{(C:N_{bulk} - 3.8)}{C:N_{bulk}} \right),$$

where $_{norm}$ was the lipid-normalized $\delta^{13}C$ value, and $_{bulk}$ were the unadjusted $\delta^{13}C$ values and C : N ratios. The same equation was applied to samples of *M. alfredi* faeces, given the zooplankton-based diet of this species. Values of $\delta^{13}C$ for seagrass samples were not adjusted.

## 2.5. Statistical analyses

One-way ANOVAs were used to investigate the effect of extraction treatment type, sampling year, sex and life stage on the values of $\delta^{15}N$ and $\delta^{13}C$ and the ratio of C : N in *M. alfredi* muscle tissue. Tukey's honestly significant difference *post hoc* tests were used to examine group-specific values when significant differences were observed. Differences between groups were assessed using non-parametric Kruskal–Wallis (KW) tests, where data were shown to be non-normally distributed using Shapiro–Wilk normality tests, or heterogeneous in nature through Levene's tests. Dunn tests were then used to examine group-specific differences *post hoc*. Similarly, linear models were used to examine the effect of wingspan on values of $\delta^{15}N$ and $\delta^{13}C$ and the ratio of C : N in *M. alfredi* muscle tissue, and of size on the same values and ratio for the muscle tissue of reef fishes. Where data were found to be non-normally distributed, Spearman's ranked-order correlation coefficients were used. All analyses were conducted in R (version 3.4.1; R Core Team 2017) and variation around the mean presented as standard deviation unless otherwise stated. Significance for all analyses was $p < 0.05$.

The packages *SIBER* [53] and *nicheROVER* [54] were used to assess the level of trophic niche overlap that occurred between male and female *M. alfredi* in each sampling year as described by Shipley *et al.* [55]. Values of $\delta^{13}C$ and $\delta^{15}N$ for both sexes were compared using a bi-plot, and the total area of the convex hull (TA) and core trophic niche area with a small sample size correction ($SEA_c$) for each sex was calculated using *SIBER*. Total trophic overlap values for 95% TA were calculated using *nicheROVER*. This latter analysis is insensitive to sample size and incorporates a statistical uncertainty using

Bayesian methods that differ from more traditional, geometric-based computations regarding trophic niche space [55,56].

Estimates of relative trophic position (TL) for *M. alfredi* were calculated using the following equation [25,57]:

$$\text{TL} = \left( \frac{(\delta^{15}\text{N}_{\text{consumer}} - \delta^{15}\text{N}_{\text{base}})}{\text{DTDF}} \right) + 2,$$

where $\delta^{15}\text{N}_{\text{consumer}}$ is the $\delta^{15}\text{N}$ value for *M. alfredi*, and $\delta^{15}\text{N}_{\text{base}}$ represents the weighted average value of $\delta^{15}\text{N}$ for all pelagic and emergent zooplankton samples combined. The integer value of 2 was used to reflect the baseline trophic level of the zooplankton samples that were composed predominantly of primary consumers (TL = 2) [26,29]. To account for the sensitivity of TL estimations to assumptions regarding the trophic fractionation of $\delta^{15}\text{N}$, two estimates of TL were generated for *M. alfredi* using diet-tissue discrimination factors (DTDFs) calculated previously for other elasmobranch species [29]. The first estimate was based upon a DTDF of 2.3‰ as calculated for *Carcharias taurus* and *Negaprion brevirostris* muscle [58], and the second based upon a DTDF of 3.7‰ as calculated for muscle of *Triakis semifasciata* [59].

Estimates of trophic enrichment between *M. alfredi* and both pelagic and emergent zooplankton samples were calculated using the following equation [16]:

$$\Delta X = \delta X_{\text{predator}} - \delta X_{\text{prey}},$$

where $X$ represents either $\delta^{15}\text{N}$ or $\delta^{13}\text{C}$, and using both bulk and lipid-normalized $\delta^{13}\text{C}$ values.

Lastly, Bayesian stable isotope mixing models were constructed to determine the potential contribution of different prey sources to the diet of *M. alfredi* in 2017 using the *simmr* package [60] in R. Other studies of the trophic ecology of both *M. alfredi* and *M. birostris* have suggested that demersal and/or mesopelagic organisms may form a key component of their diet [16,29,33]. As we could not sample mesopelagic zooplankton in this analysis, we included stable isotope data from four species of small (35.3–69.3 mm total length) mesopelagic fishes (*Ceratoscopelus warmingii*, n = 20; *Diaphus splendidus*, n = 15; *Notoscopelus caudispinosus*, n = 13; *Vinciguerra nimbaria*, n = 8) that were collected within the Indian South Subtropical Gyre for another study [61]. Each of the four species have distributions encompassing the Seychelles region and equivalent trophic positions to that of primary and secondary copepod consumers (TL = 2.1–2.9) [62]. Mesopelagic fishes have been shown to display strong isotopic similarity to mesopelagic zooplankton [63] allowing their use as representative mesopelagic organisms in the absence of zooplankton samples [29]. The selected species had an overall mean $\delta^{15}\text{N}$ value of 8.23 ± 1.27‰ and overall mean $\delta^{13}\text{C}$ value of −18.33 ± 0.39‰.

The final mixing models included three potential prey sources for *M. alfredi*; pelagic and emergent zooplankton, and mesopelagic prey sources. Data for pelagic and emergent zooplankton collected at D'Arros Island were not pooled as there were significant differences between their values of $\delta^{15}\text{N}$ and $\delta^{13}\text{C}$. We constructed two versions of our mixing models to overcome uncertainty in the importance of dietary lipid content and in the lipid normalization procedures applied to the zooplankton data. The first model used the non-normalized values of $\delta^{13}\text{C}$ for the zooplankton groups, and the second, the lipid-normalized values of $\delta^{13}\text{C}$. Both models incorporated the DTDFs of Couturier *et al.* [16] for *M. alfredi*—1.3‰ for $\delta^{13}\text{C}$ and 2.4‰ for $\delta^{15}\text{N}$—and accounted for the variation that has been observed in measurements of these values in elasmobranchs during laboratory-based experiments by introducing a standard deviation of 1‰ for both isotopes [33,58,59]. The average between the two mixing models was taken as the final estimated contribution of the three prey sources to the diet of *M. alfredi*.

# 3. Results

## 3.1. Extraction treatment effects

Lipid and urea extraction had a significant effect on values of $\delta^{15}\text{N}$ and $\delta^{13}\text{C}$, and the ratios of C : N of *M. alfredi* muscle tissue (Kruskal–Wallis test, $H(3) = 11.983$, $p = 0.007$; Kruskal–Wallis test, $H(3) = 10.440$, $p = 0.015$; ANOVA, $F_{3,16} = 220.601$, $p < 0.001$). Untreated muscle tissue samples were found to have significantly lower $\delta^{15}\text{N}$ values than the DIW, LE and LE + DIW treatment groups, which did not differ significantly from one another (figure 2*a*). The untreated control group was found to have similar values of $\delta^{13}\text{C}$ to that of the LE and LE + DIW treatments, but these values were significantly

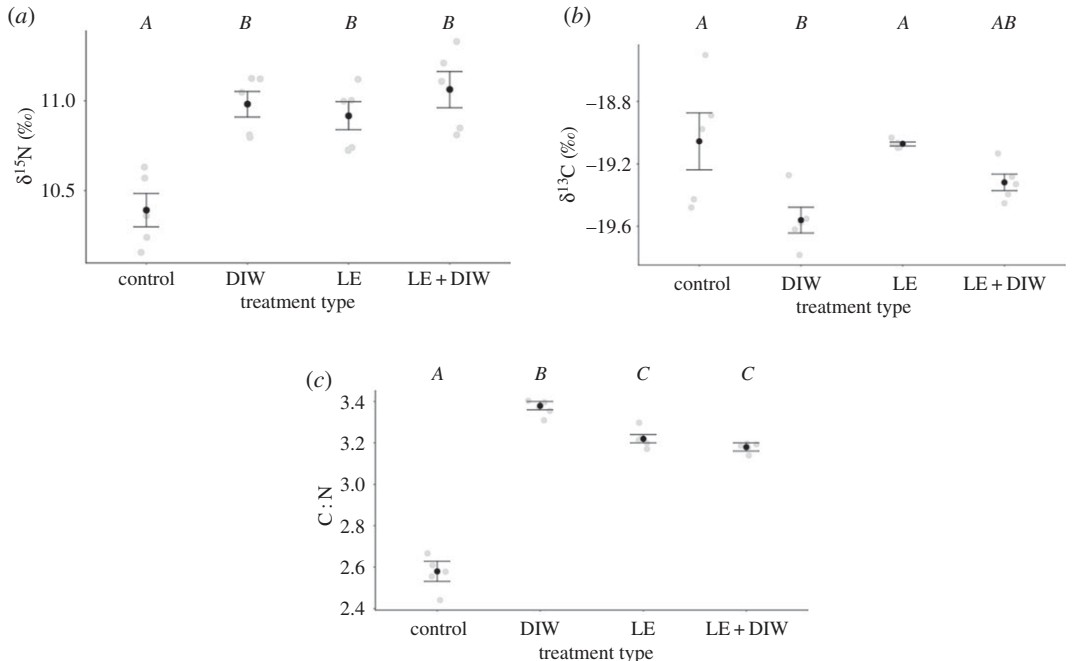

**Figure 2.** Comparison of urea and lipid extraction treatment effects on the mean values ± s.e. of $\delta^{15}$N (*a*), $\delta^{13}$C (*b*) and C : N (*c*) ratios of reef manta ray (*Mobula alfredi*) muscle tissue. Treatments with different letters are significantly different (*p* < 0.05). Control, untreated; DIW, urea extraction only; LE, lipid extraction only; LE + DIW, lipid and urea extraction.

higher than the $\delta^{13}$C values of the DIW treatment. Values of $\delta^{13}$C for the DIW and LE + DIW treatments did not differ significantly (figure 2*b*). Values of $\delta^{13}$C encompassed an overall range of 1.09‰. The C : N ratios of the untreated muscle tissue samples of *M. alfredi* were significantly lower than those of the DIW, LE and LE + DIW treatment groups. Ratios of C : N did not differ between the LE and LE + DIW treatments, but these two treatments had ratios that were significantly lower than the DIW treatment and higher than the control (figure 2*c*). Ratios of C : N for the control and DIW treatments differed significantly.

## 3.2. Stable isotopes

### 3.2.1. Reef manta rays

The effect of sampling year, sex, life stage and wingspan on isotope composition for *M. alfredi* was investigated using stable isotope data collected from the set of 50 samples included in the LE + DIW treatment (electronic supplementary material, table S2). Values of $\delta^{15}$N and $\delta^{13}$C differed significantly among *M. alfredi* muscle tissues collected in November 2016 (*n* = 13) and 2017 (*n* = 37). Values of $\delta^{15}$N were lower in 2016 than in 2017, whereas values of $\delta^{13}$C were more enriched. Consequently, ratios of C : N were lower in 2016 than in 2017 (Kruskal–Wallis tests, $H(1) = 12.321$, $p < 0.001$; $H(1) = 14.821$, $p < 0.001$; $H(1) = 12.162$, $p < 0.001$; figure 3), and all subsequent analyses considered isotope data relative to year of collection.

Values of $\delta^{15}$N and $\delta^{13}$C and ratios of C : N did not differ significantly between males (wingspan 2.1–3.6 m) and females (wingspan 2.4–3.8 m) in 2016 or in 2017 (ANOVA, $F_{1,11} = 0.991$, $p = 0.341$; $F_{1,11} = 0.153$, $p = 0.904$; $F_{1,11} = 0.002$, $p = 0.967$; ANOVA, $F_{1,34} = 3.456$, $p = 0.072$; Kruskal–Wallis test, $H(1) = 0.146$, $p = 0.702$; Kruskal–Wallis test, $H(1) = 0.584$, $p = 0.445$). Males, however, displayed lower amounts of variation in all values than females (electronic supplementary material, table S2).

The similarity of $\delta^{15}$N and $\delta^{13}$C values for male and female *M. alfredi* was confirmed by trophic overlap analyses. The trophic niche of females overlapped with 71.6% and 51.6% of that of males in November 2016 and 2017, respectively, whereas the trophic niche of males overlapped with 78.1% and 89.3% of the niche of females in November 2016 and 2017, respectively (electronic supplementary material, figure S1). Females were found to have higher TA and $SEA_c$ values in comparison to males during both sampling years, with the exception of $SEA_c$ in 2016, which was slightly lower (electronic supplementary material, table S3).

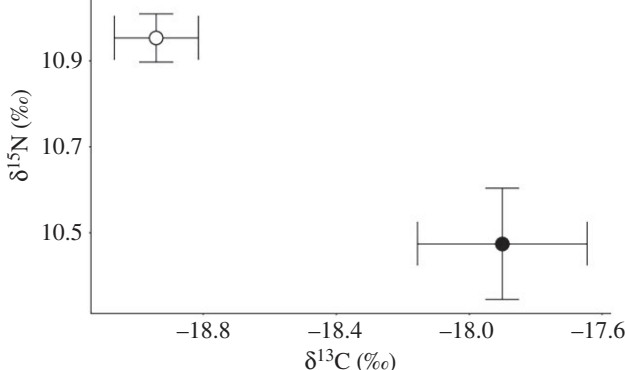

**Figure 3.** Isoscape of $\delta^{15}N$ and $\delta^{13}C$ values for reef manta ray (*Mobula alfredi*) muscle tissue collected in 2016 (black, $n = 13$) and 2017 (white, $n = 36$) at D'Arros Island, Seychelles.

Life stage did not influence values of $\delta^{15}N$, $\delta^{13}C$ or ratios of C : N for *M. alfredi* in 2016 or 2017 ($p > 0.05$), whereas wingspan was found only to influence values of $\delta^{15}N$ in 2017 (Spearman rank correlation, $r_s = -0.334$, $p = 0.047$). In November 2017, $\delta^{15}N$ tended to be lower for individuals with larger wingspans; however, these values did not vary to the extent that the C : N ratio for *M. alfredi* also differed during this sampling year.

### 3.2.2. Reef fishes

Of the 20 fish species sampled in this study, significant relationships between size and values of $\delta^{15}N$, $\delta^{13}C$ and ratios of C : N were identified across seven species (table 1). Values of $\delta^{15}N$ increased significantly with size in three species: *Aethaloperca rogaa*, *Scarus rubroviolaceus* and *Thunnus albacares* (linear regression, $r^2 = 0.591$, $F_{1,6} = 8.667$, $p = 0.026$; $r^2 = 0.525$, $F_{1,7} = 7.731$, $p = 0.027$; $r^2 = 0.673$, $F_{1,6} = 12.38$, $p = 0.013$). Similarly, values of $\delta^{13}C$ increased significantly with size in four species: *Chaetodon trifisciatus*, *Chlorurus sordidus*, *Lenthrinus lentjan* and *Selar crumenophthalmus* (linear regression, $r^2 = 0.739$, $F_{1,8} = 22.619$, $p = 0.001$; $r^2 = 0.527$, $F_{1,8} = 8.925$, $p = 0.017$; $r^2 = 0.408$, $F_{1,8} = 9.248$, $p = 0.047$; $r^2 = 0.564$, $F_{1,8} = 10.353$, $p = 0.012$, respectively). Ratios of C : N increased with size in two species i.e. *Lethrinus lentjan* and *Lutjanus bohar* (linear regression, $r^2 = 0.536$, $F_{1,8} = 9.248$, $p = 0.016$; Spearman rank correlation, $r_s = 0.906$, $p < 0.001$).

### 3.2.3. Zooplankton

Pelagic samples from net tows were dominated by calanoid copepods (figure 4*a*), with one sample showing an abundance of fish eggs (figure 4*b*) and another of crab zoea (figure 4*c*). Emergent samples were more variable in composition, with dominant species across samples including ostracods, polychaetes and crab megalopae (figure 4*d*–*f*, respectively).

Values of $\delta^{15}N$ and $\delta^{13}C$ for pelagic zooplankton samples differed between collection years. Values of $\delta^{15}N$ were larger in 2016 than in 2017 ($8.439 \pm 0.401$ and $7.185 \pm 0.340$‰, respectively; ANOVA, $F_{1,15} = 48.727$, $p < 0.001$). Values of $\delta^{13}C$ were more enriched in 2017 than in 2016 ($-21.410 \pm 1.560$ and $-22.476 \pm 0.380$‰, respectively; Kruskal–Wallis test, $H(1) = 9.482$, $p = 0.002$). Ratios of C : N did not differ significantly between years (Kruskal–Wallis test, $H(1) = 1.120$, $p = 0.290$).

Furthermore, values of $\delta^{15}N$ and $\delta^{13}C$ also differed between samples of pelagic and emergent zooplankton collected in November 2017 (ANOVA, $F_{1,14} = 87.746$, $p < 0.001$; Kruskal–Wallis test, $H(1) = 7.868$, $p = 0.005$, respectively), with the former having higher concentrations of both isotopes than the latter (table 1). Ratios of C : N were similar between these groups ($p > 0.05$).

## 3.3. Feeding ecology of reef manta rays at D'Arros Island

The relative trophic position of all taxa was visualized using an isoscape comparing the values of $\delta^{13}C$ to those of $\delta^{15}N$ (figure 5). The trophic level of *M. alfredi* was 2.92 (i.e. TL = ~3; secondary consumer) [26], after estimates using the conservative (2.3‰; TL = 3.13) and maximum (3.7‰; TL = 2.70) DTDF were averaged. The sample of *M. alfredi* faecal tissue closely aligned with that of the pelagic zooplankton samples. Muscle tissue samples of *M. alfredi* were positioned in a unique isotopic space within the

**Table 1.** Summary of values of $\delta^{15}N$, $\delta^{13}C$ and C : N (mean ± s.d.) ratios reported for reef manta ray (*Mobula alfredi*), reef fishes, zooplankton and seagrass sampled at D'Arros Island and St. Joseph Atoll, Seychelles. Values in italics indicate the presence of a significant relationship between isotope composition and size in fishes (fork length, cm).

| sample type | trophic guild | species | n | $\delta^{15}N$ (‰) | $\delta^{13}C$ (‰) | C : N |
|---|---|---|---|---|---|---|
| *Mobula alfredi* | planktivore | *Mobula alfredi* (tissue) | 50 | 10.82 ± 0.42 | −18.64 ± 0.93 | 3.13 ± 0.16 |
| | | *Mobula alfredi* (faeces)[a] | 1 | 7.88 | −21.95 | 2.79 |
| reef fishes | herbivore | *Chlorurus sordidus* | 10 | 9.24 ± 0.35 | −14.62 ± 1.32 | 3.20 ± 0.08 |
| | | *Scarus rubroviolaceus* | 9 | 9.74 ± 0.25 | −17.04 ± 1.14 | 3.16 ± 0.03 |
| | detritivore | *Crenimugil crenilabis* | 10 | 8.17 ± 1.05 | −10.70 ± 0.53 | 3.19 ± 0.03 |
| | planktivore | *Caesio teres* | 10 | 11.90 ± 0.18 | −19.17 ± 0.42 | 3.24 ± 0.08 |
| | | *Caesio xanthonota* | 10 | 11.56 ± 0.13 | −19.65 ± 0.45 | 3.30 ± 0.19 |
| | | *Pterocaesio tile* | 1 | 11.37 | −19.57 | 3.13 |
| | corallivore | *Chaetodon trifasciatus* | 10 | 10.69 ± 0.33 | −15.23 ± 0.65 | 3.21 ± 0.03 |
| | invertivore | *Lethrinus enigmaticus* | 7 | 13.42 ± 0.43 | −17.75 ± 1.04 | 3.34 ± 0.18 |
| | | *Lethrinus lentjan* | 10 | 13.19 ± 0.40 | −18.09 ± 0.91 | 3.34 ± 0.26 |
| | | *Lethrinus nebulosus* | 10 | 12.32 ± 0.65 | −16.71 ± 2.32 | 3.30 ± 0.17 |
| | | *Parupeneus macronemus* | 10 | 11.65 ± 0.92 | −17.85 ± 1.95 | 3.21 ± 0.07 |
| | reef carnivore | *Cephalopholis sonnerati* | 2 | 13.51 ± 0.21 | −19.24 ± 0.00 | 3.21 ± 0.01 |
| | | *Variola louti* | 10 | 13.50 ± 0.23 | −18.91 ± 0.34 | 3.23 ± 0.05 |
| | reef piscivore | *Aethaloperca rogaa* | 8 | 13.17 ± 0.38 | −18.31 ± 0.50 | 3.20 ± 0.03 |
| | | *Cephalopholis miniata* | 1 | 12.66 | −19.03 | 3.16 |
| | | *Lutjanus bohar* | 10 | 12.98 ± 0.34 | −18.59 ± 0.72 | 3.25 ± 0.06 |
| | reef and pelagic piscivore | *Selar crumenophthalmus* | 10 | 12.23 ± 0.30 | −19.44 ± 0.36 | 3.17 ± 0.02 |
| | pelagic piscivore | *Katsuwonus pelamis* | 5 | 12.66 ± 0.22 | −19.53 ± 0.50 | 3.16 ± 0.07 |
| | | *Sarda orientalis* | 1 | 12.86 | −19.46 | 3.54 |
| | | *Thunnus albacares* | 8 | 12.28 ± 0.25 | −19.50 ± 0.41 | 3.10 ± 0.02 |
| zooplankton | | *pelagic*[a] | 17 | 7.78 ± 0.74 | −20.92 ± 1.97 | 2.70 ± 0.27 |
| | | *emergent*[a] | 7 | 9.29 ± 0.56 | −16.79 ± 1.51 | 1.82 ± 0.22 |
| seagrass | | *Thalassodendron ciliatum* | 10 | 2.27 ± 0.79 | −10.78 ± 1.13 | 24.27 ± 2.47 |

[a]Lipid-adjusted $\delta^{13}C$ values.

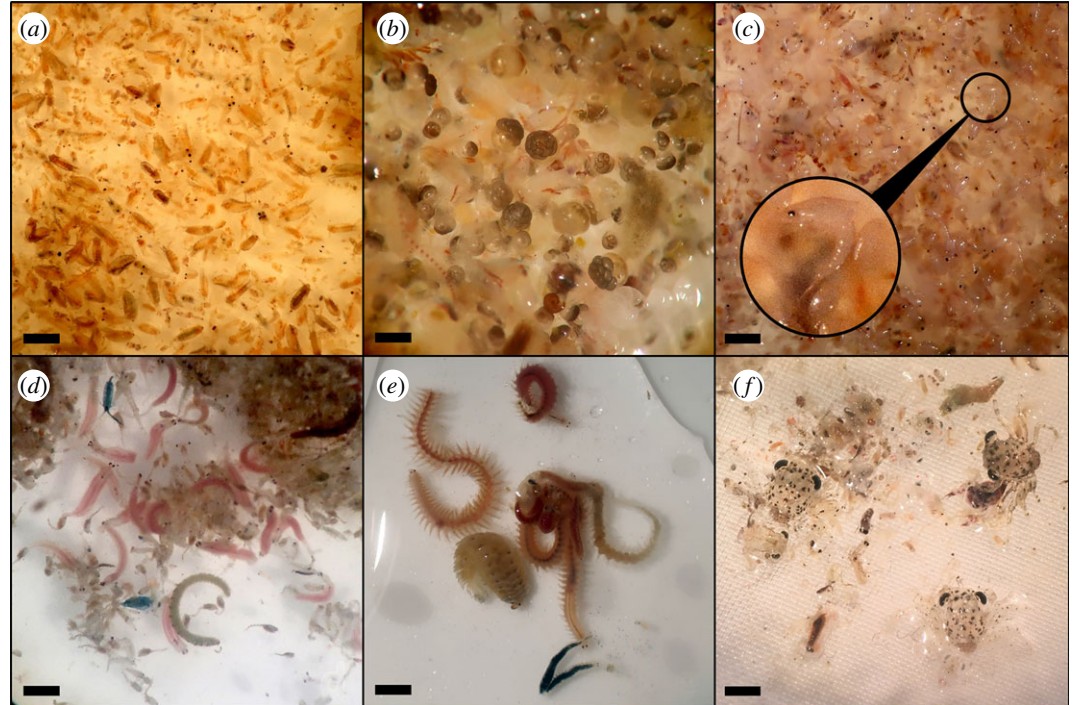

**Figure 4.** Photographs of representative pelagic (*a–c*) and emergent (*d–f*) zooplankton samples collected at D'Arros Island, Seychelles. Pelagic samples were collected during daylight hours using a towed plankton net and were dominated by copepods (*a*), fish eggs (*b*) and crab zoea (*c*). Emergent samples were collected using a light trap deployed after sunset for 2.5 h and were dominated by polychaetes and ostracods (*d*), polychaete worms (*e*) and crab megalopae (*f*). Scale bar (approx. sizes; *a–f*): 1, 0.5, 1, 2, 2, 2 mm.

isoscape, aligning closely with the zooplanktivorous and mesopelagic fishes, as well as emergent zooplankton. Values of $\delta^{13}$C for muscle tissue samples of *M. alfredi* were more enriched than those of the other zooplanktivorous fishes sampled in this study, and were similar to those of the benthic invertivore, *Parupeneus macronemus*.

Average enrichment values between *M. alfredi* and bulk emergent zooplankton samples was 1.53‰ and −0.39‰ for $\delta^{15}$N and $\delta^{13}$C, respectively. When lipids were mathematically normalized for emergent samples, the $\delta^{13}$C enrichment value decreased to −1.85‰. For bulk pelagic zooplankton samples, average enrichment values for *M. alfredi* were 3.05‰ and 3.27‰ for $\delta^{15}$N and $\delta^{13}$C, respectively. Following mathematical lipid normalization, the value of $\delta^{13}$C decreased to 2.28‰. For both mixing models, pelagic zooplankton was the dominant contributor (around 45%) to the diet of *M. alfredi* (table 2) and emergent zooplankton (approx. 38%) contributed a larger proportion of the diet than mesopelagic sources (approx. 17%; figure 6).

## 4. Discussion

*Mobula alfredi* fed predominantly on pelagic zooplankton that accumulated at the surface of the water column (approx. 50% of diet). Emergent and mesopelagic zooplankton contributed a smaller, but significant proportion of the diet (38 and 17%, respectively). The observed pattern of foraging was consistent between sexes and the majority of individuals, and placed *M. alfredi* within a unique trophic niche relative to the other reef fishes sampled at D'Arros Island.

### 4.1. Feeding ecology of reef manta rays

Mean nitrogen enrichment values indicated that *M. alfredi* at D'Arros Island occupy a trophic level of approximately 3, a level representative of a secondary consumer as was expected for this zooplanktivore [16,26,64]. The average enrichment value for $\delta^{15}$N of 2.92‰ was close to that of 2.4‰ calculated for *M. alfredi* on the Great Barrier Reef [16] and places *M. alfredi* within the range of DTDFs

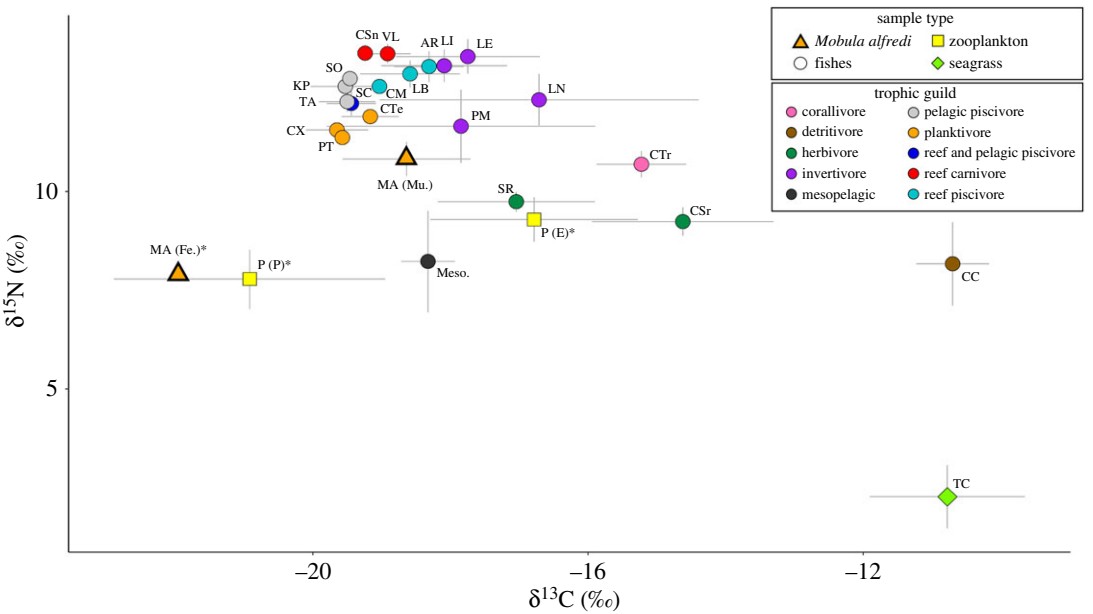

**Figure 5.** Mean values ± s.d. of δ$^{13}$C and δ$^{15}$N for samples of reef manta ray (*Mobula alfredi*), 20 species of fishes, zooplankton and seagrass collected at D'Arros Island and St. Joseph Atoll, Seychelles. Representative mean values of mesopelagic fishes (Meso.) sampled in the Western Indian Ocean are also included. Figure legend outlines allocation of symbols to sample types and colours to trophic guild of sampled fish species. Asterisks represent lipid-normalized δ$^{13}$C values. *Mobula alfredi*: faeces, MA (Fe.)*; muscle, MA (Mu.). Fishes: *Aethaloperca rogaa*, AR; *Caesio teres*, CTe; *Caesio xanthonota*, CX; *Cephalopholis miniata*, CM; *Cephalopholis sonnerati*, CSn; *Chaetodon trifasciatus*, CTr; *Chlorurus sordidus*, CSr; *Crenimugil crenilabis*, CC; *Katsuwonus pelamis*, KP; *Lethrinus enigmaticus*, LE; *Lethrinus lentjan*, LL; *Lethrinus nebulosus*, LN; *Lutjanus bohar*, LB; Mesopelagic, Meso.; *Parupeneus macronemus*, PM; *Pterocaesio tile*, PT; *Sarda orientalis*, SO; *Scarus rubroviolaceus*, SR; *Selar crumenophthalmus*, SC; *Thunnus albacares*, TA; *Variola louti*, VL. Zooplankton: emergent, P (E)*; pelagic, P (P)*. Seagrass: *Thalassodendron ciliatum*, TC.

**Table 2.** Outputs of mixing models estimating the proportional contribution (±s.d.) of pelagic and emergent zooplankton and mesopelagic sources to the diet of reef manta rays (*Mobula alfredi*) at D'Arros Island, Seychelles, based on samples collected in November 2017. Model 1 used bulk δ$^{13}$C values of zooplankton samples, whereas Model 2 used values mathematically normalized for lipids. Both models assumed a diet-tissue discrimination factor (DTDF) of 2.3 ± 1.0‰. The mean values were calculated as an average between the two mixing models.

| source | Model 1 | Model 2 | Mean |
|---|---|---|---|
| emergent zooplankton | 0.41 ± 0.06 | 0.34 ± 0.07 | *0.38 ± 0.07* |
| pelagic zooplankton | 0.43 ± 0.04 | 0.47 ± 0.05 | *0.45 ± 0.05* |
| mesopelagic sources | 0.16 ± 0.06 | 0.19 ± 0.08 | *0.17 ± 0.07* |
| s.d. δ$^{13}$C | 0.26 ± 0.20 | 0.75 ± 0.48 | *0.50 ± 0.34* |
| s.d. δ$^{15}$N | 0.13 ± 0.10 | 0.16 ± 0.12 | *0.14 ± 0.11* |

estimated for other elasmobranchs (2.29‰ for *C. taurus* and *N. brevirostris*, and 3.7‰ for *T. semifasciata*) [58,59,65].

The values of δ$^{13}$C in *M. alfredi* muscle tissues after lipid and urea extraction (LE + DIW) fell between those of the lipid-normalized pelagic and emergent zooplankton groups with enrichment values of −2.28‰ and −1.85‰, respectively. Isotopic signatures of carbon for *M. alfredi* were similar to those of planktivorous fishes, a benthic invertivore (*P. macronemus*) and mesopelagic zooplankton. These results suggest that *M. alfredi* may have periods of residency (often several months) in reef environments, followed by shorter excursions between reefs into the open ocean and movements throughout the water column [18,19,37].

Values of δ$^{15}$N and δ$^{13}$C for *M. alfredi* muscle tissue were significantly different between collection years, likely reflecting a shift in the zooplankton community structure at D'Arros Island during this

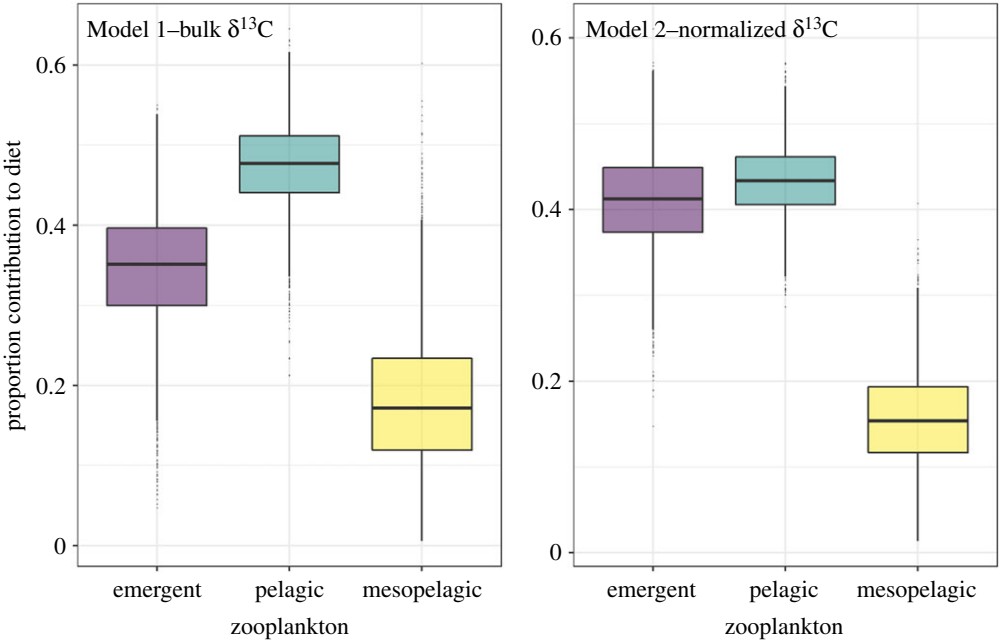

**Figure 6.** Estimated contribution of emergent and pelagic zooplankton, and mesopelagic sources, to the diet of the reef manta ray (*Mobula alfredi*) at D'Arros Island and St. Joseph Atoll, Seychelles, in 2017 (purple, blue and yellow, respectively). Proportions estimated using two Bayesian stable isotope mixing models that assumed a diet-tissue discrimination factor (DTDF) of 2.3 ± 1.0‰ and included either bulk (Model 1) or lipid-normalized (Model 2) $\delta^{13}C$ values. The central box spans the 2.5–97.5% confidence intervals and the middle line denotes the median. Note variable scale on *y*-axis.

time and subsequently, prey availability to *M. alfredi* [66,67]. Within each year, values of $\delta^{15}N$ and $\delta^{13}C$ did not vary between the sexes, although females tended to display more variation in both isotopic concentrations. This suggests that female *M. alfredi* may forage upon a more diverse assemblage of prey items than males. In 2017, the values of $\delta^{15}N$ for *M. alfredi* were found to decrease with increasing wingspan. Given that the level of residency displayed by *M. alfredi* to D'Arros Island decreases with increasing wingspan [37], it is possible that this result reflects the tendency for larger individuals (mostly females) to seek alternative foraging opportunities offshore of D'Arros Island and consume zooplankton communities that occupy lower trophic positions [67].

## 4.2. Zooplankton as prey for reef manta rays

Pelagic zooplankton were found to constitute the largest proportion (45%) of the diet of *M. alfredi* at D'Arros Island. The coral reefs surrounding these locations are one of the few locations in Seychelles where *M. alfredi* are known to aggregate predictably year-round [37], and individuals are regularly sighted surface-feeding during daylight hours. This is particularly true within the St. Joseph Channel (figure 1), where pelagic zooplankton is observed to accumulate along current lines (L. Peel 2018, unpublished data). The isotopic similarity between faeces of *M. alfredi* and pelagic zooplankton reported here confirms that *M. alfredi* are consuming these communities at D'Arros Island, and estimates from faeces are likely representative of a very recent feeding event (possibly within 24 h of sampling). Similar observations of *M. alfredi* feeding on pelagic zooplankton at the surface during the day been reported at other aggregation sites around the world [12,14,20,68–70]. *Mobula alfredi* have also been observed foraging during the day on demersal zooplankton in the Maldives [70], and on the Great Barrier Reef. In the latter locality, demersal zooplankton constitute a significant proportion of the diet [16]. Such benthic-oriented feeding behaviours have, however, not been observed for *M. alfredi* at D'Arros Island during the day (L. Peel 2018, unpublished data). Taken together, the observations reported here and the results of the isotopic mixing models suggest that pelagic zooplankton comprises the majority of the diet of *M. alfredi* at D'Arros Island.

The feeding behaviour of *M. alfredi* on pelagic zooplankton in the St. Joseph Channel may serve to increase nutrient cycling over the reefs of D'Arros Island as a whole, and enhance nutrient enrichment in particular places within the reef. After feeding on zooplankton, *M. alfredi* frequently return to

cleaning stations on coral reefs for parasite removal, to socialize, and possibly to thermoregulate [11,43,71]. Defecation is often observed at these shallow (less than 30 m) cleaning stations (electronic supplementary material, figure S2), where the water temperature can be warmer than the surrounding habitat, possibly increasing the speed of digestion [70,72]. At D'Arros Island, this could result in the transfer of nutrients from St. Joseph Channel to the reef slope, with positive impacts on coral growth. Similar impacts of nutrient enrichment on coral growth have been reported for reef fishes and sharks [4,21,73,74], and for schools of small planktivorous fishes that shelter in individual coral heads [75]. The high residency and frequent use of cleaning stations by *M. alfredi* at reefs around D'Arros Island identified through acoustic telemetry [37] and photo-identification (L. Peel 2018, unpublished data), increases the significance of such nutrient cycling processes to these reefs.

Emergent zooplankton were estimated to comprise approximately 38% of the diet of *M. alfredi*, suggesting that the species—unlike diurnal zooplanktivorous fishes [34]—also forages along the reef at D'Arros Island at night. The value of $\delta^{13}C$ for *M. alfredi* was close to that of the invertivore, *P. macronemus*, which forages on benthic invertebrates in sand along reef edges during the day (less than 40 m) [76,77]. This apparently contradictory result can be explained by the fact that the same communities consumed by benthic invertivores during the day emerge at dusk to occupy the water column, where they can become prey for *M. alfredi*. While it is possible that emergent zooplankton originating from the St. Joseph Atoll lagoon may also contribute slightly to this demersal signature, acoustic telemetry and visual observations indicate that *M. alfredi* rarely enter this habitat [37]. In contrast to the feeding behaviour of *M. alfredi* within the lagoon at Palmyra Atoll [17], this suggests that foraging by *M. alfredi* on the emergent zooplankton community at D'Arros Island is restricted to the reefs surrounding this location.

The emergence of benthic zooplankton from sediment and the reef at night is thought to influence the movement and foraging behaviour of *M. alfredi* in many locations, including eastern Australia [11,16], Hawaii [78], Indonesia [13] and Seychelles [37]. The ability to forage throughout a full diel cycle may be necessary in order for these relatively large animals to obtain sufficient food resources to satisfy metabolic requirements in the warm surface waters of a coral reef [79]. In contrast to *M. alfredi*, smaller planktivorous fishes tend to be either diurnal (e.g. caesionids, pomacentrids) or nocturnal (e.g. holocentrids, apogonids), and subsequently only forage during half of the day. The extended periods of foraging undertaken by *M. alfredi* may, therefore, serve to increase nutrient cycling in the reef environment over larger temporal scales, particularly at cleaning sites within the reef where nutrient supply is expected to be enhanced.

In addition to nutrient cycling within reef systems, the possible contribution of mesopelagic zooplankton to the diet of *M. alfredi* (approx. 17%) highlights the potential for this species to act as a vector for horizontal nutrient transport between coastal and mesopelagic ecosystems. In comparison to the local-scale nutrient supply occurring across the reefs at D'Arros Island (less than 1 km), the transport of nutrients derived from mesopelagic origins is anticipated to occur over larger distances (greater than 10 km). Mesopelagic zooplankton communities perform diel vertical migrations involving movements from depths (greater than 200 m) during the day to shallow surface waters (less than 50 m) during the night [80]. Feeding on these communities would require travel by *M. alfredi* either 10 km to the east or 20 km to the west of D'Arros Island to waters beyond the shelf edge. Such distances fall well below the maximum reported range of travel for *M. alfredi* within a 24 h period (89.3 km d$^{-1}$) [37], and would be reduced should the mesopelagic zooplankton community migrate horizontally over the Amirantes Bank and towards D'Arros Island during the night [81]. Although the frequency at which such ventures occur cannot be assessed here, such behaviour could provide *M. alfredi* with additional feeding opportunities, or supplement foraging when food availability around D'Arros Island is scarce [18,82,83]. The ability of *M. alfredi* to travel away from the shallow reef of D'Arros Island to consume mesopelagic zooplankton contrasts with zooplanktivorous reef fishes, which tend to range over a much smaller spatial scale (metre to kilometre). The excretion of faecal matter by *M. alfredi* on return to D'Arros Island may therefore represent a unique method of nutrient supply to the coral reefs at this locality, and may serve to increase horizontal nutrient transport in the region [4,17,21].

## 4.3. Extraction procedures for reef manta ray muscle tissue

There is still debate regarding the most appropriate way to treat elasmobranch tissue samples prior to analysis, despite the increasing use of stable isotopes as a means of investigating the feeding ecology of marine megafauna and the trophic structure of marine ecosystems [22]. We found that urea should

be extracted at a minimum from *M. alfredi* tissue samples prior to stable isotope analysis, given the significant difference observed in all $\delta^{15}N$ treatment groups relative to the control. Lipid extraction must also be considered in studies of *M. alfredi* stable isotope analyses (as discussed by Marcus *et al.* [47]), as significant differences were observed in values of $\delta^{13}C$ across extraction procedures. No difference was observed between the final C : N ratio of the LE and LE + DIW extraction procedures, suggesting that the lipid extraction process conducted here was sufficient for the concurrent removal of lipids and urea from *M. alfredi* muscle tissues [45]. Future studies of *M. alfredi* using stable isotope analyses should, therefore, aim to extract lipids from freeze-dried muscle tissue samples using the LE procedure described above. No additional urea extraction treatment would then be required, and the utilization of a consistent extraction methodology would increase the comparability of results of all studies.

## 4.4. Limitations

All sampling was conducted during the month of November in both 2016 and 2017, which has implications for our ability to detect seasonal shifts in the diet of *M. alfredi*, and align these measures with biological and environmental variables. White muscle tissue is estimated to represent the assimilated diet of elasmobranchs over periods of 300–700 days [65,84], in contrast to shorter periods reflected by tissues such as blood plasma or skin, which exhibit higher turnover rates of $\delta^{15}N$ and $\delta^{13}C$ [59,85,86]. The restriction of our sampling to a single month of the year at D'Arros Island thus averages our view of the feeding ecology of *M. alfredi* on the zooplankton community at D'Arros Island across both the southeast (April–September) and northwest (December–March) monsoonal periods. Future studies should aim to collect tissue samples throughout the year in order to gain further insight into potential seasonal shifts in the foraging and feeding patterns of *M. alfredi* in Seychelles throughout the full annual cycle. Additionally, other tissues, such as skin [87] or mucus [88], with higher turnover rates of isotopes could be collected and analysed together with samples of muscle tissue to provide better temporal resolution of feeding patterns. Such tissue samples would also provide more information on the frequency and timing of mesopelagic foraging, and whether pelagic zooplankton remains the primary food source for *M. alfredi* year-round.

Significant differences in the $\delta^{15}N$ and $\delta^{13}C$ values of pelagic zooplankton collected in 2016 and 2017 further emphasize the importance of multi-year sampling programmes for studies of manta ray feeding ecology. These differences are likely to be the result of a dynamic zooplankton community existing at D'Arros Island, the composition of which may be constantly changing on both spatial and temporal scales [66,67]. While logistical constraints limited the amount of zooplankton sampling that could be completed in the present study, future research should endeavour to repeatedly sample zooplankton communities throughout the annual cycle. Concurrently, rapid methods for community structure assessment (e.g. using ZooScan systems and Plankton ID software) [14] should be developed to allow for the feeding ecology and trophic role of *M. alfredi* to be described on finer spatio-temporal scales at aggregation sites around the world.

## 5. Conclusion

Stable isotope analyses revealed that *M. alfredi* occupy a unique trophic niche and role within the coral reef ecosystem at D'Arros Island. With a diet that includes pelagic, emergent and mesopelagic zooplankton, *M. alfredi* are able to maximize foraging opportunities at this remote locality while supplying nutrients via excretion to the coral reef over fine (less than 1 km) and broad (greater than 10 km) spatial scales. The high level of site fidelity that *M. alfredi* exhibits at D'Arros Island [37] increases the significance of these nutrient transport processes from mesopelagic ecosystems to local reefs by increasing the frequency at which they can occur. Collectively, the findings presented here highlight the potential for this large elasmobranch to play a unique role in nutrient transport and supply to an otherwise nutrient-poor coral reef ecosystem.

Ethics. All research was approved by the Seychelles Bureau of Standards, Seychelles Ministry of Environment and Energy and The University of Western Australia (RA/3/100/1480). Relevant CITES, import and export permits were obtained, where required.
Data accessibility. A summary of sampled reef fishes, stable isotope values for sexes and life stage classes of *M. alfredi*, and results of trophic overlap analyses are provided in the electronic supplementary material.

Authors' contributions. L.R.P., R.D., G.M.W.S. and M.G.M. conceived of the study. L.R.P. and S.P.C. assisted with permit acquisitions. L.R.P., R.D., C.A.K.D. and G.M.W.S. conducted the fieldwork. L.R.P. analysed the data and created figures. L.R.P., R.D. and M.G.M. drafted the manuscript. All authors provided feedback on the manuscript and gave final approval for publication.

Competing interests. The authors declare that they have no conflict of interest.

Funding. This study was funded by the Save Our Seas Foundation (SOSF). It was also supported by the Manta Trust, the Australian Institute of Marine Science and a Robson & Robertson Award from The University of Western Australia.

Acknowledgements. This study was made possible with support from the SOSF-DRC. We thank G. Skrzypek and the West Australian Biogeochemistry Centre for assistance with the stable isotope analyses. Additionally, we thank L. Gordon for his invaluable assistance in the field, and S. Hollanda at the Seychelles Fishing Authority for overseeing the freeze-drying procedures of all collected samples. Finally, we thank the editors and two anonymous reviewers for their valuable feedback on this manuscript.

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
