## [Reviewer comments · Royal Society Open Science]

Review History

RSOS-190599.R0 (Original submission)

Review form: Reviewer 1

Is the manuscript scientifically sound in its present form?

Yes

Are the interpretations and conclusions justified by the results?

Yes

Is the language acceptable?

Yes

Is it clear how to access all supporting data?

Yes

Do you have any ethical concerns with this paper?

No

Have you any concerns about statistical analyses in this paper?

No

Recommendation?

Major revision is needed (please make suggestions in comments)

Comments to the Author(s)

The authors use stable isotopes to determine which prey reef manta rays consume and relate this to their ability to act as nutrient vectors. They make some important contributions to our understanding of manta ray foraging, however, they need to provide clarity in terms of what this means regarding manta foraging ecology. Furthermore, without any provided movement data, some of their conclusion regarding nutrient transfer are tenuous.

Line 44: nutrient transfer is not only through defecation but also excretion. In fact nutrients excreted are in a form more easily utilized by primary producers and may be of greater importance.

Line 52-53: The authors should make clear in the introduction the distinction between pelagic plankton, emergent, and mesopelagic plankton. In the intro it should be made clear how they differ and where (and when) they are found. I found myself confused as these distinctions were not made.

Line 53-56: the combination of oceanic and reef manta rays here is confusing. Why is *Birostris* mentioned in line 55? Furthermore, these results do not consider those from the central Pacific where reef mantas spend the majority of their time in an atoll lagoon, which is where they get most of their nutrients. The central Pacific results do not suggest manta rays are major offshore vectors (see McCauley et al. 2015, Marine Biology).

Line 123-129: The pelagic plankton I found very confusing. Where were they collected from? The channel? The lagoon or foreef? It's not clear. How does the surface plankton signal compare between different parts of the atoll? Was this considered?

Line 174-175: I assume this correction is for zooplankton samples but this is never explicitly stated.

Line 203: The TL estimates assume zooplankton as the base of the food chain. However, technically shouldn't it be for phytoplankton?

Line 224-228: I do find it problematic that the mesopelagic plankton signal is taken from mesopelagic fish, and it's not clear from where they were sampled. It might be better to have someone with greater expertise in mixing models review this section.

Fig 5: Why are there no variability estimates for manta ray results? I see no error bars.

Line 330-333: The authors found manta muscle tissues to be different for those samples in 2017 and 2018. However, they combine both for the mixing models. It's not clear to me how this won't confound the results as the variability could give a false measure of foraging across sources (as opposed to potential variability in baseline). Please clarify.

Line 407-408: warm waters may speed up digestion but not necessarily 'aid'

Line 402-414: the interpretation of these results are completely contingent on an understanding of movements. For example, how do they know surface foraging of pelagic plankton didn't occur in the lagoon, or along the forereef? With the long turnover time of tissues it seems very hard to interpret results in terms of nutrient vectors, without movement data. I see the authors have a tracking paper in review and it would be much more convincing if that paper were published so that the movements are known. I really like the idea of mantas creating nutrient hotspots at cleaning stations (irrespective of where nutrients come from), but the argument for them connecting ecosystems is less convincing as it stands.

Line 426-431: Again here we have results from long temporal time frames (isotopes with tissue turnover times up to a year) being used to make inference of behavior over diel time scales (hours). Without some telemetry results to back up, these conclusions sound speculative.

Line 436-437: How would mantas be transferring nutrients over such long time scales? After eating a meal, it would only be a time period of days before all digesta has been voided?

Line 443-448: As presented, there is not data to make this claim.

The conclusion doesn't mention other aspects of the results: a) why were there differences in manta values between the two years? and b) why were males more enriched in N than females?

Review form: Reviewer 2 (Andrew Fischer)

Is the manuscript scientifically sound in its present form?

Yes

Are the interpretations and conclusions justified by the results?

Yes

Is the language acceptable?

Yes

Is it clear how to access all supporting data?

Yes

Do you have any ethical concerns with this paper?

No

Have you any concerns about statistical analyses in this paper?

No

Recommendation?

Accept with minor revision (please list in comments)

Comments to the Author(s)

Excellent work in all aspects. The only potential issue would be sampling restricted to one month. This limitation has been identified. Additional research to examine differences between monsoonal and non-monsoonal periods would be beneficial and important to link to other research. The readability of the sentence on Ines 318-320 might be improved if a comma was added after "[23]."

Where are the results of the generalised linear model? Can you provide some more details on how the models were set up? Assumptions? Data distribution?

A more detailed title that reflects the outcomes of the research would be useful, unless length is limited by the journal.

Were any environmental variables collected in this study? Presumably, these could also impact the results which were limited to 2 years and one month. How would you environmental changes produce different results? This would be nice to see in the discussion.

Decision letter (RSOS-190599.R0)

03-Jul-2019

Dear Ms Peel,

The editors assigned to your paper ("Trophic ecology of the reef manta ray (*Mobula alfredi*) at a remote coral reef") have now received comments from reviewers. We would like you to revise your paper in accordance with the referee and Associate Editor suggestions which can be found below (not including confidential reports to the Editor). Please note this decision does not guarantee eventual acceptance.

Please submit a copy of your revised paper before 26-Jul-2019. Please note that the revision deadline will expire at 00.00am on this date. If we do not hear from you within this time then it will be assumed that the paper has been withdrawn. In exceptional circumstances, extensions may be possible if agreed with the Editorial Office in advance. We do not allow multiple rounds of revision so we urge you to make every effort to fully address all of the comments at this stage. If deemed necessary by the Editors, your manuscript will be sent back to one or more of the original reviewers for assessment. If the original reviewers are not available, we may invite new reviewers.

- Data accessibility

If you wish to submit your supporting data or code to Dryad (<http://datadryad.org/>), or modify your current submission to dryad, please use the following link:
<http://datadryad.org/submit?journalID=RSOS&manu=RSOS-190599>

- **Competing interests**

- **Authors' contributions**

- **Acknowledgements**

- **Funding statement**

Kind regards,

Alice Power

Editorial Coordinator

on behalf of Dr Asha de Vos (Associate Editor) and Kevin Padian (Subject Editor)
openscience@royalsociety.org

Associate Editor's comments (Dr Asha de Vos):

Please consider incorporating and/or responding to the revisions requested by the reviewers.
Thank you.

Subject Editor Comments to Author:

Although we're checking "major revision," it appears that most of the changes requested are minor ones. Please attend to these and let us know that you've responded to them. We look forward to your revision.

Comments to Author:

Reviewers' Comments to Author:

Reviewer: 1

Comments to the Author(s)

The authors use stable isotopes to determine which prey reef manta rays consume and relate this to their ability to act as nutrient vectors. They make some important contributions to our understanding of manta ray foraging, however, they need to provide clarity in terms of what this means regarding manta foraging ecology. Furthermore, without any provided movement data, some of their conclusion regarding nutrient transfer are tenuous.

Line 44: nutrient transfer is not only through defecation but also excretion. In fact nutrients excreted are in a form more easily utilized by primary producers and may be of greater importance.

Line 52-53: The authors should make clear in the introduction the distinction between pelagic plankton, emergent, and mesopelagic plankton. In the intro it should be made clear how they differ and where (and when) they are found. I found myself confused as these distinctions were not made.

Line 53-56: the combination of oceanic and reef manta rays here is confusing. Why is *Birostris* mentioned in line 55? Furthermore, these results do not consider those from the central Pacific where reef mantas spend the majority of their time in an atoll lagoon, which is where they get most of their nutrients. The central Pacific results do not suggest manta rays are major offshore vectors (see McCauley et al. 2015, Marine Biology).

Line 123-129: The pelagic plankton I found very confusing. Where were they collected from? The channel? The lagoon or foreef? It's not clear. How does the surface plankton signal compare between different parts of the atoll? Was this considered?

Line 174-175: I assume this correction is for zooplankton samples but this is never explicitly stated.

Line 203: The TL estimates assume zooplankton as the base of the food chain. However, technically shouldn't it be for phytoplankton?

Line 224-228: I do find it problematic that the mesopelagic plankton signal is taken from mesopelagic fish, and it's not clear from where they were sampled. It might be better to have someone with greater expertise in mixing models review this section.

Fig 5: Why are there no variability estimates for manta ray results? I see no error bars.

Line 330-333: The authors found manta muscle tissues to be different for those samples in 2017 and 2018. However, they combine both for the mixing models. It's not clear to me how this won't confound the results as the variability could give a false measure of foraging across sources (as opposed to potential variability in baseline). Please clarify.

Line 407-408: warm waters may speed up digestion but not necessarily 'aid'

Line 402-414: the interpretation of these results are completely contingent on an understanding of movements. For example, how do they know surface foraging of pelagic plankton didn't occur in the lagoon, or along the forereef? With the long turnover time of tissues it seems very hard to interpret results in terms of nutrient vectors, without movement data. I see the authors have a tracking paper in review and it would be much more convincing if that paper were published so that the movements are known. I really like the idea of mantas creating nutrient hotspots at

cleaning stations (irrespective of where nutrients come from), but the argument for them connecting ecosystems is less convincing as it stands.

Line 426-431: Again here we have results from long temporal time frames (isotopes with tissue turnover times up to a year) being used to make inference of behavior over diel time scales (hours). Without some telemetry results to back up, these conclusions sound speculative.

Line 436-437: How would mantas be transferring nutrients over such long time scales? After eating a meal, it would only be a time period of days before all digesta has been voided?

Line 443-448: As presented, there is not data to make this claim.

The conclusion doesn't mention other aspects of the results: a) why were there differences in manta values between the two years? and b) why were males more enriched in N than females?

Reviewer: 2

Comments to the Author(s)

Excellent work in all aspects. The only potential issue would be sampling restricted to one month. This limitation has been identified. Additional research to examine differences between monsoonal and non-monsoonal periods would be beneficial and important to link to other research. The readability of the sentence on Ines 318-320 might be improved if a comma was added after "[23]."

Where are the results of the generalised linear model? Can you provide some more details on how the models were set up? Assumptions? Data distribution?

A more detailed title that reflects the outcomes of the research would be useful, unless length is limited by the journal.

Were any environmental variables collected in this study? Presumably, these could also impact the results which were limited to 2 years and one month. How would you environmental changes produce different results? This would be nice to see in the discussion.

Author's Response to Decision Letter for (RSOS-190599.R0)

See Appendix A.

Decision letter (RSOS-190599.R1)

08-Aug-2019

Dear Ms Peel,

I am pleased to inform you that your manuscript entitled "Stable isotope analyses reveal unique trophic role of reef manta rays (*Mobula alfredi*) at a remote coral reef" is now accepted for publication in Royal Society Open Science.

Kind regards,

on behalf of Dr Asha de Vos (Associate Editor) and Kevin Padian (Subject Editor)
openscience@royalsociety.org

Associate Editor Comments to Author (Dr Asha de Vos):

Great work and thanks for addressing all the revisions. I think the paper is ready!

Appendix A

Response Letter

Trophic ecology of the reef manta ray (*Mobula alfredi*) at a remote coral reef

[RSOS-190599]

Associate & Subject Editor - Comments

Comment	Response
Associate Editor Please consider incorporating and/or responding to the revisions requested by the reviewers. Thank you. Subject Editor Although we're checking "major revision," it appears that most of the changes requested are minor ones. Please attend to these and let us know that you've responded to them. We look forward to your revision.	We thank the editors and reviewers for their comments on this manuscript. We have incorporated and responded to all requests, and have updated the manuscript where required. Please note that the line numbers referenced in this document relate to the tracked changes version of the updated manuscript, including the supplementary material.

Reviewer #1 - Comments

Comment	Response
Overview: The authors use stable isotopes to determine which prey reef manta rays consume and relate this to their ability to act as nutrient vectors. They make some important contributions to our understanding of manta ray foraging, however, they need to provide clarity in terms of what this means regarding manta foraging ecology. Furthermore, without any provided movement data, some of their conclusion regarding nutrient transfer are tenuous.	We thank the reviewer for this comment. Our acoustic telemetry study was recently published [reference #37 in-text] and we have made reference to the movement data provided within it throughout the updated manuscript. Notably: LINES 564-567: "Given that the level of residency displayed by M. alfredi to D'Arros Island decreases with increasing wingspan [37], it is possible that this result reflects the tendency for larger individuals (mostly females) to seek alternative foraging opportunities offshore of D'Arros Island and St. Joseph Atoll and consume zooplankton communities that occupy lower trophic positions" LINES 596-599: "The high residency and frequent use of cleaning stations by M. alfredi at reefs around D'Arros Island identified through acoustic telemetry [37] and photo-identification (L. Peel, unpub. data), increases the significance of such nutrient cycling processes to these reefs." LINES 614-619: "While it is possible that emergent zooplankton originating from the St. Joseph Atoll lagoon may also contribute slightly to this demersal signature, acoustic telemetry and visual observations indicate that M. alfredi rarely enter this habitat [37]. In contrast to the feeding behaviour of M. alfredi within the lagoon at Palmyra Atoll [17], this suggests that foraging by M. alfredi on the emergent zooplankton community at D'Arros Island and St. Joseph Atoll is restricted to the reefs surrounding both land masses."

	LINES 637-641: “Feeding on these communities would require travel by M. alfredi either 10 km to the east or 20 km to the west of D’Arros Island waters beyond the shelf edge. Such distances fall well below the maximum reported range of travel for M. alfredi within a 24 hour period (89.3 km d⁻¹) [37], and would be reduced should the mesopelagic zooplankton community migrate horizontally over the Amirantes Bank and towards D’Arros Island during the night [81].” The reference information has been included at LINE 955: [37] Peel, L.R., Stevens, G.M., Daly, R., Daly, C.A.K., Lea, J.S., Clarke, C.R., Collin, S.P. and Meekan, M.G., 2019. Movement and residency patterns of reef manta rays Mobula alfredi in the Amirante Islands, Seychelles. Marine Ecology Progress Series, 621, pp.169-184.
Line 44: nutrient transfer is not only through defecation but also excretion. In fact nutrients excreted are in a form more easily utilized by primary producers and may be of greater importance.	We thank the reviewer for acknowledging this accidental oversight. This section of the introduction has been updated to read “...where they may excrete waste products” at LINE 77.
Line 52-53: The authors should make clear in the introduction the distinction between pelagic plankton, emergent, and mesopelagic plankton. In the intro it should be made clear how they differ and where (and when) they are found. I found myself confused as these distinctions were not made.	We thank the reviewer for this comment. The second paragraph of the introduction has been updated to include these definitions. It reads as follows on LINES 68-79: “Reef manta rays (Mobula alfredi) [8, 9] are large filter-feeding elasmobranchs that display high levels of site fidelity and residency at circum-tropical reef locations [10-13]. Individuals are often observed feeding on pelagic zooplankton that accumulates near the surface of the water column (< 5 m) during daylight hours, and this foraging behaviour has been found to be linked to zooplankton density in eastern Australia [14]. During the night, demersal zooplankton emerge from the seabed, where they vertically migrate towards the surface [15], and become potential prey items for M. alfredi [16]. These emergent zooplankton communities are thought to be particularly significant for M. alfredi that occupy lagoon systems [17]. Furthermore, M. alfredi have been observed to travel offshore to feed on mesopelagic zooplankton before returning to inshore coral reefs during the day where they may excrete waste products [16, 18-20]. In this way, M. alfredi may be able to create links between shallow coral reefs and deeper water ecosystems, potentially facilitating the horizontal transport of nutrients between these environments [21].”
Line 53-56: the combination of oceanic and reef manta rays here is confusing. Why is birostris mentioned in line 55? Furthermore, these results do not consider those from the central Pacific where reef mantas spend the majority of their time in an atoll lagoon, which is where they get most of their nutrients. The central Pacific results do not suggest manta rays are major offshore vectors (see McCauley et al. 2015, Marine Biology).	Oceanic manta rays (Mobula birostris) were included here as published stable isotope studies for manta rays are currently very limited. This was one of the few examples available in the literature that presented $\delta^{13}\text{C}$ and $\delta^{15}\text{N}$ data. The study completed in Palmyra Atoll focussed upon $\delta^{34}\text{S}$ and $\delta^{15}\text{N}$, and was not originally cited for consistency. This section of the introduction has since been updated clarify the mention of M. birostris and to include the findings from the central Pacific at LINES 85-107 as follows: “Few studies to-date have used this analytical approach to examine the feeding ecology of both reef and oceanic (Mobula birostris) manta rays, but those published have shown that emergent ($\delta^{13}\text{C} > -17\text{‰}$) [27] and mesopelagic zooplankton (> 200 m depth in the water column) comprise a significant proportion of the diet of M. alfredi along the coast of eastern Australia [16], and of M. birostris in Ecuador [29], respectively. Emergent zooplankton have also been reported to be a significant contributor to the diet of M. alfredi within the lagoon of Palmyra Atoll in the central Pacific [17]. Coupled with the potential for

	M. alfredi to travel large distances (> 300 km) [19, 30, 31], these findings suggest that manta rays may act as a vector for the horizontal transport of nutrients between offshore and coastal reef ecosystems. Additionally, the high site fidelity displayed by M. alfredi at aggregation sites may serve to increase the significance of such nutrient transfer processes, and of the trophic role of this species within reef environments as a whole.”
Line 123-129: The pelagic plankton I found very confusing. Where were they collected from? The channel? The lagoon or foreef? It’s not clear. How does the surface plankton signal compare between different parts of the atoll? Was this considered?	All samples were collected over the reef flats of D’Arros Island. No samples were collected from within the St. Joseph Atoll lagoon as M. alfredi have not been observed to enter this site. Sea grass samples represented the base of the food web within the lagoon. These methodologies have also been clarified in-text at LINES 179-184 as follows: “During the day, zooplankton samples were collected within the uppermost two metres of the water column using a small plankton net towed behind an 18 ft research boat. The net (202 µm mesh, 50 cm diameter; General Oceanics, FL, USA) was deployed when M. alfredi were sighted feeding over the reef flats of D’Arros Island or the along the reef edge of the St. Joseph Channel during November 2016 (n = 7) and 2017 (n = 10), and towed for approximately five minutes at a speed of two knots”
Line 174-175: I assume this correction is for zooplankton samples but this is never explicitly stated.	The use of corrections for the zooplankton samples has been clarified at LINES 238-239, which now read: “No corrections were required for M. alfredi or any of the reef fishes, but zooplankton $\delta^{13}\text{C}$ values were normalised with the following equation [29, 52]”
Line 203: The TL estimates assume zooplankton as the base of the food chain. However, technically shouldn’t it be for phytoplankton?	We thank the reviewer for this question. The value of ‘2’ in the estimates for TL reflects accounts for zooplankton being primary consumers (TL = 2), rather than primary producers (TL = 1). As it was not possible to sample phytoplankton at D’Arros Island, zooplankton were used to estimate the TL of M. alfredi.
Line 224-228: I do find it problematic that the mesopelagic plankton signal is taken from mesopelagic fish, and it’s not clear from where they were sampled. It might be better to have someone with greater expertise in mixing models review this section.	Mesopelagic fishes have been shown to reflect a similar trophic position to mesopelagic zooplankton (Valls et al. 2014). The protocol for integrating these data into mixing models is also presented in Burgess et al (2016) for oceanic manta rays.
Fig 5: Why are there no variability estimates for manta ray results? I see no error bars.	Figure 5 originally presented mean values \pm S.E, and the error bars for the manta rays were too small to be seen. This figure has now been updated at LINE 482 to present mean \pm S.D. values to allow for increased comparability with Table 2. Error bars for reef manta ray samples are now visible as a result.
Line 330-333: The authors found manta muscle tissues to be different for those samples in 2017 and 2018. However, they combine both for the mixing models. It’s not clear to me how this won’t confound the results as the variability could give a false measure of foraging across sources (as opposed to potential variability in baseline). Please clarify.	We thank the reviewer for this comment. The stable isotope data has now been re-analysed to consider the manta ray and plankton samples collected in different years separately. The final mixing model now contains only data from samples collected in 2017 (when both pelagic and emergent zooplankton samples were collected). Details of these updated analyses have been provided throughout the manuscript as follows: Abstract LINES 28-48: “The isotopic signatures of nitrogen ($\delta^{15}\text{N}$) and carbon ($\delta^{13}\text{C}$) for M. alfredi differed by year, but did not vary by sex or life stage, suggesting that all individuals occupy the same trophic niche at this coral reef. Furthermore, the

isotopic signatures for *M. alfredi* differed to those for co-occurring planktivorous fish species also sampled at D'Arros Island and St. Joseph Atoll, suggesting that the ecological niche of *M. alfredi* is unique. Pelagic zooplankton were the main contributor (45%) to the diet of *M. alfredi*, combined with emergent zooplankton (38%) and mesopelagic prey items (17%). Given the extent of movement that would be required to undertake this foraging strategy, individual *M. alfredi* are implicated as important vectors of nutrient supply around and to the coral reefs surrounding D'Arros Island and St. Joseph Atoll, particularly where substantial site fidelity is displayed by these large elasmobranchs."

Methods

LINES 245-246: "One-way ANOVAs were used to investigate the effect of extraction treatment type, sampling year, sex, and life stage on the values of $\delta^{15}\text{N}$ and $\delta^{13}\text{C}$ and the ratio of C:N in *M. alfredi* muscle tissue."

LINES 263-272: "The packages *SIBER* [53] and *nicheROVER* [54] were used to assess the level of trophic niche overlap that occurred between male and female *M. alfredi* in each sampling year as described by Shipley et al. (2018). Values of $\delta^{13}\text{C}$ and $\delta^{15}\text{N}$ for both sexes were compared using a bi-plot, and the total area of the convex hull (TA) and core trophic niche area with a small sample size correction (SEAc) for each sex was calculated using *SIBER*. Total trophic overlap values for 95% TA were calculated using *nicheROVER*."

LINES 289-291: "Lastly, Bayesian stable isotope mixing models were constructed to determine the potential contribution of different prey sources to the diet of *M. alfredi* in 2017 using the *simmr* package [60] in R."

Results

LINES 349-356: "The effect of sampling year, sex, life stage and wingspan on isotope composition was investigated using stable isotope data collected from the set of 50 samples included in the LE+DIW treatment (Table S2). Values of $\delta^{15}\text{N}$ and $\delta^{13}\text{C}$ differed significantly among *M. alfredi* muscle tissues collected in November 2016 ($n = 13$) and 2017 ($n = 37$). Values of $\delta^{15}\text{N}$ were lower in 2016 than in 2017, whereas values of $\delta^{13}\text{C}$ were more enriched. Consequently, ratios of C:N were lower in 2016 than 2017 (Kruskal-Wallis tests, $H(1) = 12.321$, $P < 0.001$; $H(1) = 14.821$, $P < 0.001$, $H(1) = 12.162$, $P < 0.001$, respectively; Fig. 3), and all subsequent analyses considered isotope data relative to year of collection."

LINES 362-395: "Values of $\delta^{15}\text{N}$ and $\delta^{13}\text{C}$ and ratios of C:N did not differ significantly between males (wingspan 2.1 – 3.6 m) and females (wingspan 2.4 m – 3.8 m) in 2016 or 2017 (ANOVA, $F(1,11) = 0.153$, $P = 0.904$; $F(1,11) = 0.991$, $P = 0.341$; $F(1,11) = 0.002$, $P < 0.967$; Kruskal-Wallis test, $H(1) = 0.146$, $P = 0.702$; ANOVA, $F(1,34) = 3.456$, $P = 0.072$; Kruskal-Wallis test, $H(1) = 0.584$, $P = 0.445$, respectively). Males, however, displayed lower amounts of variation in all values than females (Table S2).

The similarity $\delta^{15}\text{N}$ and $\delta^{13}\text{C}$ values for male and female *M. alfredi* was confirmed by trophic overlap analyses. The trophic niche of females overlapped with 71.6% and 51.6% of that of males in November 2016, and the trophic niche of males overlapped with 78.1% and 89.34% of the niche of females in November 2017 (Fig S1). Females were found to have higher TA and SEAc values in comparison to males during both sampling years, with the exception of SEAc in 2016, which was slightly lower (Table S3).

Life stage did not influence values of $\delta^{15}\text{N}$, $\delta^{13}\text{C}$, or ratios of C:N for *M. alfredi* in 2016 or 2017 ($P > 0.05$), whereas wingspan was found only to influence values of $\delta^{15}\text{N}$ in 2017. In November 2017, $\delta^{15}\text{N}$ tended to be lower for individuals with larger wingspans in 2017, however, these values did not vary to the extent that the C:N ratio for *M. alfredi* also differed during this sampling year.”

LINES 434-438: “Values of $\delta^{15}\text{N}$ and $\delta^{13}\text{C}$ for pelagic zooplankton samples differed between collection years. Values of $\delta^{15}\text{N}$ were larger in 2016 than 2017 (8.439 ± 0.401 and $7.185 \pm 0.340\text{‰}$, respectively; ANOVA, $F(1,15) = 48.727$, $P < 0.001$). Values of $\delta^{13}\text{C}$ were more enriched in 2017 than in 2016 (-21.410 ± 1.560 and $-22.476 \pm 0.380\text{‰}$, respectively; Kruskal-Wallis test, $H(1) = 9.482$, $P = 0.002$). Ratios of C:N did not differ significantly between years (Kruskal-Wallis test, $H(1) = 1.120$, $P = 0.290$).

Furthermore, values of $\delta^{15}\text{N}$ and $\delta^{13}\text{C}$ also differed between samples of pelagic and emergent zooplankton collected in November 2017 (ANOVA, $F(1,14) = 87.746$, $P < 0.001$; Kruskal-Wallis test, $H(1) = 7.868$, $P = 0.005$, respectively), with the former having higher concentrations of both isotopes than the latter (Table 1). Ratios of C:N were similar between these groups ($P > 0.05$).”

LINES 467-470: “For both mixing models, pelagic zooplankton was the dominant contributor (around 45%) to the diet of *M. alfredi* (Table 2) and emergent zooplankton (~38%) contributed a larger proportion of the diet than mesopelagic sources (~17%) (Fig. 6).”

Additionally, the results presented in Table 2, Figure 6, Table S2, Table S3 and Figure S1 have been updated to reflect the grouping of samples by collection year.

Discussion

LINES 539-541: “*M. alfredi* fed predominantly on pelagic zooplankton that accumulated at the surface of the water column (approximately 50% of diet). Emergent and mesopelagic zooplankton contributed a smaller, but significant proportion of the diet (38 and 17%, respectively)”

LINES 558-567: “Values of $\delta^{15}\text{N}$ and $\delta^{13}\text{C}$ for *M. alfredi* muscle tissue were significantly different between collection years, likely reflecting a shift in the zooplankton community structure at D’Arros Island during this time and subsequently, prey availability to *M. alfredi* [66, 67]. Within each year, values of $\delta^{15}\text{N}$ and $\delta^{13}\text{C}$ did not vary between the sexes, although females tended to display more variation in both isotopic concentrations. This suggests that female *M. alfredi* may forage upon a more diverse assemblage of prey items than males. In 2017, the values of $\delta^{15}\text{N}$ for *M. alfredi* were found to decrease with increasing wingspan. Given that the level of residency displayed by *M. alfredi* to D’Arros Island decreases with increasing wingspan [37], it is possible that this result reflects the tendency for larger individuals (mostly females) to seek alternative foraging opportunities offshore of D’Arros Island and St. Joseph Atoll and consume zooplankton communities that occupy lower trophic positions [67].”

LINES: 698-707: “Significant differences in the $\delta^{15}\text{N}$ and $\delta^{13}\text{C}$ values of pelagic zooplankton collected in 2016 and 2017 further emphasise the importance of multi-year sampling programs for studies of manta ray feeding ecology. These differences are likely to be the result of a dynamic zooplankton community existing at D’Arros Island and St. Joseph Atoll, the composition of which may be constantly changing on both spatial and temporal scales [66, 67]. While logistical constraints limited the amount of zooplankton sampling that could be completed

	in the present study, future research should endeavour to repeatedly sample zooplankton communities throughout the annual cycle. Concurrently, rapid methods for community structure assessment (e.g. using ZooScan systems and Plankton ID Software) [14] should be developed to allow for the feeding ecology and trophic role of M. alfredi to be described on finer spatio-temporal scales at aggregation sites around the world.”
Line 407-408: warm waters may speed up digestion but not necessarily 'aid'	The sentence at LINES 591-593 has been updated to read: “Defecation is often observed at these shallow (< 30 m) cleaning stations (Fig. S2), where the water temperature can be warmer than the surrounding habitat, possibly increasing the speed of digestion [66, 68]”
Line 402-414: the interpretation of these results are completely contingent on an understanding of movements. For example, how do they know surface foraging of pelagic plankton didn't occur in the lagoon, or along the forereef? With the long turnover time of tissues it seems very hard to interpret results in terms of nutrient vectors, without movement data. I see the authors have a tracking paper in review and it would be much more convincing if that paper were published so that the movements are known. I really like the idea of mantas creating nutrient hotspots at cleaning stations (irrespective of where nutrients come from), but the argument for them connecting ecosystems is less convincing as it stands.	As described above, detailed references to our recently published acoustic telemetry study have been included in this updated manuscript. The following additions have been made: LINES 558-567: “This suggests that female M. alfredi may forage upon a more diverse assemblage of prey items than males. In 2017, the values of $\delta^{15}\text{N}$ for M. alfredi were found to decrease with increasing wingspan. Given that the level of residency displayed by M. alfredi to D'Arros Island decreases with increasing wingspan [37], it is possible that this result reflects the tendency for larger individuals (mostly females) to seek alternative foraging opportunities offshore of D'Arros Island and St. Joseph Atoll and consume zooplankton communities that occupy lower trophic positions [67].” LINES 614-619: “While it is possible that emergent zooplankton originating from the St. Joseph Atoll lagoon may also contribute slightly to this demersal signature, acoustic telemetry and visual observations indicate that M. alfredi rarely enter this habitat [37]. In contrast to the feeding behaviour of M. alfredi within the lagoon at Palmyra Atoll [17], this suggests that foraging by M. alfredi on the emergent zooplankton community at D'Arros Island and St. Joseph Atoll is restricted to the reefs surrounding both land masses.” LINES 637-641: “Feeding on these communities would require travel by M. alfredi either 10 km to the east or 20 km to the west of D'Arros Island waters beyond the shelf edge. Such distances fall well below the maximum reported range of travel for M. alfredi within a 24 hour period (89.3 km d⁻¹) [37], and would be reduced should the mesopelagic zooplankton community migrate horizontally over the Amirantes Bank and towards D'Arros Island during the night [81].”
Line 426-431: Again here we have results from long temporal time frames (isotopes with tissue turnover times up to a year) being used to make inference of behavior over diel time scales (hours). Without some telemetry results to back up, these conclusions sound speculative.	We thank the reviewer for this comment. We believe the updated text in the discussion (see above) has addressed this concern.
Line 436-437: How would mantas be transferring nutrients over such long time scales? After eating a meal, it would only be a time period of days before all digesta has been voided?	We thank the reviewer for this question. The text at LINES 632-641 has been updated to read as follows: “In comparison to the local-scale nutrient supply occurring across the reefs at D'Arros Island (< 1 km), the transport of nutrients derived from mesopelagic origins is anticipated to occur over larger distances (> 10 km). Mesopelagic zooplankton communities perform diel vertical migrations involving movements

	from depths (> 200 m) during the day to shallow surface waters (< 50 m) during the night [80]. Feeding on these communities would require travel by M. alfredi either 10 km to the east or 20 km to the west of D'Arros Island waters beyond the shelf edge. Such distances fall well below the maximum reported range of travel for M. alfredi within a 24 hour period (89.3 km d ⁻¹) [37], and would be reduced should the mesopelagic zooplankton community migrate horizontally over the Amirantes Bank and towards D'Arros Island during the night [81]"
Line 443-448: As presented, there is not data to make this claim.	As outlined above, we believe that the additional references to the recently published acoustic telemetry address this concern regarding the movement patterns of reef manta rays at D'Arros Island.
The conclusion doesn't mention other aspects of the results: a) why were there differences in manta values between the two years? and b) why were males more enriched in N than females?	We thank the reviewer for acknowledging this accidental oversight. The discussion has been amended to reflect the findings of the updated analyses. LINES 558-567: "Values of $\delta^{15}\text{N}$ and $\delta^{13}\text{C}$ for M. alfredi muscle tissue were significantly different between collection years, likely reflecting a shift in the zooplankton community structure at D'Arros Island during this time and subsequently, prey availability to M. alfredi [66, 67]. Within each year, values of $\delta^{15}\text{N}$ and $\delta^{13}\text{C}$ did not vary between the sexes, although females tended to display more variation in both isotopic concentrations. This suggests that female M. alfredi may forage upon a more diverse assemblage of prey items than males. In 2017, the values of $\delta^{15}\text{N}$ for M. alfredi were found to decrease with increasing wingspan. Given that the level of residency displayed by M. alfredi to D'Arros Island decreases with increasing wingspan [37], it is possible that this result reflects the tendency for larger individuals (mostly females) to seek alternative foraging opportunities offshore of D'Arros Island and St. Joseph Atoll and consume zooplankton communities that occupy lower trophic positions [67]." LINES 698-707: "Significant differences in the $\delta^{15}\text{N}$ and $\delta^{13}\text{C}$ values of pelagic zooplankton collected in 2016 and 2017 further emphasise the importance of multi-year sampling programs for studies of manta ray feeding ecology. These differences are likely to be the result of a dynamic zooplankton community existing at D'Arros Island and St. Joseph Atoll, the composition of which may be constantly changing on both spatial and temporal scales [66, 67]. While logistical constraints limited the amount of zooplankton sampling that could be completed in the present study, future research should endeavour to repeatedly sample zooplankton communities throughout the annual cycle. Concurrently, rapid methods for community structure assessment (e.g. using ZooScan systems and Plankton ID Software) [14] should be developed to allow for the feeding ecology and trophic role of M. alfredi to be described on finer spatio-temporal scales at aggregation sites around the world."

Reviewer #2 – Comments

Comment	Response
Overview: Excellent work in all aspects. The only potential issue would be sampling restricted to one month. This limitation has been identified. Additional research to examine differences between monsoonal and non-monsoonal periods would be	We thank the reviewer for these comments, and agree that additional research to examine differences between monsoonal and non-monsoonal periods would be valuable for this study and others.

beneficial and important to link to other research.	
The readability of the sentence on lines 318-320 might be improved if a comma was added after "[23]."	A comma has been added to this sentence at LINE 459.
Where are the results of the generalised linear model? Can you provide some more details on how the models were set up? Assumptions? Data distribution?	We thank the reviewer for these questions. Additional information regarding the linear models has been provided as follows: LINES 251-253: "Similarly, linear models were used to examine the effect of wingspan on values of $\delta^{15}\text{N}$ and $\delta^{13}\text{C}$ and the ratio of C:N in M. alfredi muscle tissue, and of size on the same values and ratio for the muscle tissue of reef fishes." LINES 391-395: "Life stage did not influence values of $\delta^{15}\text{N}$, $\delta^{13}\text{C}$, or ratios of C:N for M. alfredi in 2016 or 2017 ($P > 0.05$), whereas wingspan was found only to influence values of $\delta^{15}\text{N}$ in 2017 (Spearman Rank Correlation, $r_s = -0.334$, $P = 0.047$). In November 2017, $\delta^{15}\text{N}$ tended to be lower for individuals with larger wingspans in 2017, however, these values did not vary to the extent that the C:N ratio for M. alfredi also differed during this sampling year." LINES 405-407: "Ratios of C:N increased with size in two species i.e. Lethrinus lentjan and Lutjanus bohar (Linear regression, $r^2 = 0.536$, $F_{1,8} = 9.248$, $P = 0.016$; Spearman Rank Correlation, $r_s = 0.906$, $P < 0.001$)."
A more detailed title that reflects the outcomes of the research would be useful, unless length is limited by the journal.	We thank the reviewer for this suggestion. The title of the manuscript has been updated to reflect the outcomes of the research. It now reads: "Stable isotope analyses reveal unique trophic role of reef manta rays (Mobula alfredi) at a remote coral reef" At LINES 1-2.
Were any environmental variables collected in this study? Presumably, these could also impact the results which were limited to 2 years and one month. How would you environmental changes produce different results? This would be nice to see in the discussion.	Given the short time frame over which sample collection was possible during this study and the fact that both field trips were conducted in November, environmental variables were not collected to accompany the stable isotope data. Mention of environmental variables has been included in the limitations section of the discussion alongside the suggestion of extended sampling and monitoring periods. LINES 698-707: "Significant differences in the $\delta^{15}\text{N}$ and $\delta^{13}\text{C}$ values of pelagic zooplankton collected in 2016 and 2017 further emphasise the importance of multi-year sampling programs for studies of manta ray feeding ecology. These differences are likely to be the result of a dynamic zooplankton community existing at D'Arros Island and St. Joseph Atoll, the composition of which may be constantly changing on both spatial and temporal scales [66, 67]. While logistical constraints limited the amount of zooplankton sampling that could be completed in the present study, future research should endeavour to repeatedly sample zooplankton communities throughout the annual cycle. Concurrently, rapid methods for community structure assessment (e.g. using ZooScan systems and Plankton ID Software) [14] should be developed to allow for the feeding ecology and trophic role of M. alfredi to be described on finer spatio-temporal scales at aggregation sites around the world."